# Proteomics Analysis of Tangeretin-Induced Apoptosis through Mitochondrial Dysfunction in Bladder Cancer Cells

**DOI:** 10.3390/ijms20051017

**Published:** 2019-02-26

**Authors:** Jen-Jie Lin, Chun-Chieh Huang, Yu-Li Su, Hao-Lun Luo, Nai-Lun Lee, Ming-Tse Sung, Yu-Jen Wu

**Affiliations:** 1Department of Hematology-Oncology, Kaohsiung Chang Gung Memorial Hospital and Chang Gung University, College of Medicine, Kaohsiung 83301, Taiwan; q87634@hotmail.com (J.-J.L.); yolisu@mac.com (Y.-L.S.); 2Department of Biological Technology, Meiho University, Pingtung 91202, Taiwan; 3Department of Radiation Oncology, Kaohsiung Chang Gung Memorial Hospital and Chang Gung University, College of Medicine, Kaohsiung 83301, Taiwan; cgukinace@gmail.com; 4Department of Urology, Kaohsiung Chang Gung Memorial Hospital and Chang Gung University, College of Medicine, Kaohsiung 83301, Taiwan; alesy1980@gmail.com (H.-L.L.); nailun0110@gmail.com (N.-L.L.); 5Department of Pathology, Kaohsiung Chang Gung Memorial Hospital and Chang Gung University, College of Medicine, Kaohsiung 83301, Taiwan; 6Department of Nursing, Meiho University, Pingtung 91202, Taiwan

**Keywords:** tangeretin, proteomics, apoptosis, mitochondrial dysfunction

## Abstract

Tangeretin is one of the most abundant compounds in citrus peel, and studies have shown that it possesses anti-oxidant and anti-cancer properties. However, no study has been conducted on bladder cancer cells. Bladder cancer has the second highest mortality rate among urological cancers and is the fifth most common malignancy in the world. Currently, combination chemotherapy is the most common approach by which to treat patients with bladder cancer, and thus identifying more effective chemotherapeutic agents that can be safely administered to patients is a very important research issue. Therefore, this study investigated whether tangeretin can induce apoptosis and identified the signaling pathways of tangeretin-induced apoptosis in human bladder cancer cells using two-dimensional gel electrophoresis (2DGE). The results of the study demonstrated that 60 μM tangeretin reduced the cell survival of a BFTC-905 bladder carcinoma cell line by 42%, and induced early and late apoptosis in the cells. In this study 2DGE proteomics technology identified 41 proteins that were differentially-expressed in tangeretin-treated cells, and subsequently LC–MS/MS analysis was performed to identify the proteins. Based on the functions of the differentially-expressed proteins, the results suggested that tangeretin caused mitochondrial dysfunction and further induced apoptosis in bladder cancer cells. Moreover, western blotting analysis demonstrated that tangeretin treatment disturbed calcium homeostasis in the mitochondria, triggered cytochrome *C* release, and activated caspase-3 and caspase-9, which led to apoptosis. In conclusion, our results showed that tangeretin-induced apoptosis in human bladder cancer cells is mediated by mitochondrial inactivation, suggesting that tangeretin has the potential to be developed as a new drug for the treatment of bladder cancer.

## 1. Introduction

Bladder cancer ranks as the fifth most common malignancy in the world, and has the second highest mortality rate among urological cancers [1,2]. Clinical studies have revealed that transitional cell carcinoma is the most common type of urinary bladder cancer, accounting for 90% of human bladder cancers, and is one of the most common causes of death of all genitourinary cancers [3,4]. Epidemiological studies showed that several environmental factors are associated with the carcinogenesis of bladder cancer, and higher rates of bladder cancer are seen in males than in females (ratio of approximately 3:1) [5,6,7]. Currently, surgical removal of cancerous bladder tissue is the most effective initial treatment for most patients diagnosed with bladder cancer. Chemotherapy is often included as an adjuvant therapy to eliminate remaining cancer cells in the body. However, the recurrence rate of bladder cancer is high, and chemotherapy or radiotherapy is often the best treatment in patients with reoccurrence of cancer. Current chemotherapy drugs have many potential problems, including a poor specificity and drug resistance, which extensively reduce the therapeutic effect. Thus, the use of combination chemotherapy is more effective for bladder cancer treatment, as it can reduce the drug resistance of cancers. This approach has been shown to enable reductions in the doses of chemotherapy drugs and improve the prognosis [8,9]. Tangeretin is rich in the peel of sweet oranges (*Citrus sinensis)*. Previous studies have shown that flavones and flavone glycoside are abundant compounds in sweet oranges. Recent research has shown that some of the flavonoids contain multiple methoxy groups; these are known as polymethoxylated flavones (PMFs) [10,11]. This type of flavonoid displays selective cytotoxicity towards cancerous cells, but not normal cells [12], and therefore has great potential for development as a new drug for the treatment of cancer. At present, tangeretin has been shown to inhibit cell proliferation of human ovarian, rectal, gastric, breast, and prostate cancers through the induction of cell-cycle arrest and apoptosis [13,14,15,16,17,18]. Additionally, it has other biological activities, including an anti-oxidant effect, inhibition of oncogenes, and anti-angiogenesis [16,19,20,21]. The combination of tangeretin with cisplatin has been found to enhance the anti-tumor effect on human ovarian cancer cells, and reduced the dose of cisplatin required to induce cell death [14]. As no study has been performed to investigate whether tangeretin possesses bioactivity in terms of inhibition of bladder cancer growth, our present study aimed to explore the effect of tangeretin on human bladder cancer and identify the mechanisms involved.

## 2. Results

### 2.1. Cytotoxicity of Tangeretin towards Bladder Cancer Cells

Many active natural substances have been found to be cytotoxic towards cancer cells and have been clinically used for cancer treatment. Tangeretin, as shown in Figure 1A, has also been shown to have many biological activities and has been chosen for analysis of its effect against bladder cancer in vitro. We first investigated the cytotoxic activity of tangeretin against four bladder cancer cell lines, J82, BFTC-905, T24, and RT4. The disease phenotype classification of BFTC-905 and RT4 are bladder papillary epithelial cell carcinoma. J82 and T24 are bladder transitional cell carcinoma. The results showed that when the cells were treated with 20, 40, and 60 μM of tangeretin for 24 h, the BFTC-905 cells were more sensitive to the treatment than the other cell lines as shown in Figure 1B. The cell viability of the BFTC-905 cells was reduced by 42% at a concentration of 60 μM, which indicated that tangeretin has a greater cytotoxic effect on this bladder cancer cell line. As BFTC-905 cells were sensitive to tangeretin treatment, they were therefore selected as the test cells for our subsequent experiments.

### 2.2. Inhibition Effect of Tangeretin on BFTC-905 Cells

To better ascertain the cytotoxic dosage of tangeretin, we increased the tangeretin concentration to 100 μΜ, which inhibited the cell growth of BFTC-905 cells by 70%, as shown in Figure 2A. Comparison of morphological changes of cells under an inverted microscope after 24 h of tangeretin treatment with the control cells (DMSO) showed that the cell number and cell membrane shrinkage were significantly changed with an increasing concentration of tangeretin, as shown in Figure 2B. In addition to inhibition of cell growth, we performed wound-healing and transwell migration assays to examine whether tangeretin inhibited cell metastasis. In the wound-healing assay, as shown in Figure 2C, BFTC-905 cells without tangeretin treatment had significant better wound closure as compared with those treated with 60 μM tangeretin; the wound-healing ability being negatively correlated with an increasing tangeretin concentration. The transwell migration assay demonstrated that with an increased tangeretin concentration, the number of cells that invaded through the membrane decreased, as shown in Figure 2D, suggesting that tangeretin has the ability to inhibit cell migration of BFTC-905 cells, even at a low concentration.

### 2.3. Tangeretin-Induced Apoptosis in BFTC-905 Cells

In order to understand whether apoptosis is involved in the inhibition of cell proliferation in BFTC-905 bladder cancer cells by tangeretin, we utilized a fluorescent TUNEL/DAPI assay to analyze the nuclear DNA integrity. The results showed that the green fluorescent intensity was amplified with an increasing tangeretin concentration, as shown in Figure 3A, indicating that tangeretin treatment caused stress, inducing DNA fragmentation in a dose-dependent manner. Annexin V and propidium iodide (PI) labeling and flow cytometry analysis further revealed the apoptosis process. Figure 3B shows the percentages of viable (Annexin V^−^/PI^−^), early apoptotic (Annexin V^+^/PI^−^), late apoptotic (Annexin V^+^/PI^+^), and necrotic cells (Annexin V^−^/PI^+^) after tangeretin treatment. The results demonstrated that 0, 20, 40, and 60 μM tangeretin treatment caused early apoptosis in 1.3%, 6.5%, 7.66%, and 10.5%, and late apoptosis in 1.8%, 6.3%, 7.6%, and 18% of BFTC-905 cells, respectively, indicating that tangeretin caused apoptosis in bladder cancer cells, as shown in Figure 3B.

### 2.4. Use of Two-Dimensional Gel Electrophoresis to Measure Changes in Protein Expressions of BFTC-905 Cells after Tangeretin Treatment

Next, we used a two-dimensional gel electrophoresis (2DGE) proteomics approach to identify protein change in BFTC-905 cells after tangeretin treatment. The conditions used in this study were pI 4–7 and pI 3–10 NL, and separation was performed by 12.5% SDS-PAGE. The experiments were conducted in triplicate, as shown in Figure 4A–D, and the images of the gels were analyzed using PDQuest 2-D software (version 7.1.1; Bio-Rad, United States). The recognized stops were analyzed, and 41 differentially-expressed spots with at least a 1.5-fold change were identified and excised. After in-gel digestion, the samples were analyzed using LC–MS/MS to identify proteins upregulated/downregulated by tangeretin treatment. The 41 identified differentially-expressed proteins are shown in Table 1. We further performed western blotting analysis to validate the changes in these proteins caused by tangeretin treatment. As shown in Figure 4E, six representative differentially-expressed proteins (PDIA3, 14-3-3 sigma, T-complex1, Stress-70, Annexin A2, and UQCRC2) in BFTC-905 cells following tangeretin treatment (0, 20, 40, 60 μM) were analyzed, and exhibited the same patterns of change in expression as observed using 2DGE.

### 2.5. Tangeretin Induced Mitochondrial Dysfunction and Activated Caspase-Dependent Pathways in BFTC-905 Cells

The 2DGE analysis revealed that the expressions of several mitochondrial proteins were increased in tangeretin-treated BFTC-905 cells, including ATP synthase D chain, T-complex protein 1, pyruvate kinase isozymes M1/M2, triosephosphate isomerase, and ubiquinol-cytochrome-c reductase complex core protein 2. The results suggested that tangeretin affects the mitochondrial energy metabolism, which is associated with the induction of apoptosis. Several recent studies have reported that intracellular stress may trigger the intrinsic pathway of apoptosis in cells, and organelles involved in the process include mitochondria and the endoplasmic reticulum [22,23]. Mitochondrial dysfunction is known to play an important role in apoptosis. Bcl-2 family proteins play key roles in stabilizing mitochondrial function. When the homeostasis of Bcl2 proteins is disrupted, cytochrome *C* release occurs from the mitochondria to the cytosol due to an increased mitochondrial membrane potential. This further leads to activation of downstream caspase-3, caspase-6, and caspase-7, cleavage of poly (ADP-ribose) polymerase 1 (PARP-1), chromosome condensation, and DNA fragmentation. Similar pathways have been identified in bornyl *cis*-4-hydroxycinnamate, a natural active compound isolated from the *Piper betle* Linn., inducing apoptosis in melanoma cells that is associated with mitochondrial dysfunction and endoplasmic reticulum stress [24].

Subsequently, we used immunoblotting to analyze the expressions of Bcl-2 family proteins in BFTC-905 cells after treatment with different concentrations of tangeretin (0, 20, 40, 60 μM). In cells under mitochondrial stress, Bad complexes are formed with Bcl2 and Bcl-xL, leading to apoptosis, while the formation of a p-Bad complex with 14-3-3 protein can prevent apoptosis [25,26,27]. Our results showed that the expressions of Bax and Bad were increased with increasing concentrations of tangeretin and greater treatment durations, but at the same time, the expressions of Bcl2, Bcl-xL, and p-Bad were reduced. Therefore, the results suggested that tangeretin caused mitochondrial stress, resulting in an increase in Bad protein expression and reductions of Bcl2, Bcl-xL, and p-Bad protein expressions, as shown in Figure 5. The process consequently caused mitochondrial dysfunction.

Western blotting analysis also demonstrated that the expression levels of pro-caspase-9 and pro-caspase-3 gradually decreased with increasing tangeretin concentrations, while activated forms of caspases (cleaved-caspase-9 and cleaved-caspase-3) were increased and caspase-8 was unchanged. The amount of cytochrome *C* in the cytosol was also increased with increasing concentrations of tangeretin and greater treatment durations.

The release of apoptosis inducing factor (AIF) from the mitochondria through the cytosol to the nucleus can induce chromosome condensation, and activation of endonuclease G (Endo G) causes translocation from the intermembrane space of the mitochondria to the nucleus, causing DNA cleavage [28,29]. Our results showed that the expressions of AIF and Endo G were increased with increasing tangeretin concentrations, suggesting that mitochondrial stress is involved in tangeretin-induced cell death.

### 2.6. Mitochondrial Inhibitors Block Tangeretin-Induced Apoptosis

To verify that tangeretin-induced apoptosis is mediated by mitochondrial inactivation, we pre-treated BFTC-905 cells for 2 h with mitochondrial inhibitors (including trifluoperazine (TFZ), aristolochic acid (ArA), and cyclosporine A (CyA)) followed by 24 h of tangeretin treatment. We found that inhibition of a mitochondrial ion channel increased the cell viability in an MTT assay, as shown in Figure 6A, induced the expression of anti-apoptotic Bcl2 protein, and reduced the expression of pro-apoptotic Bax protein, as well as reducing the release of cytochrome *C*, as shown in Figure 6B.

The aforementioned results suggested that tangeretin induced the intrinsic pathway of apoptosis, and caused inactivation of mitochondria and changes in the mitochondrial membrane potential, leading to the destruction of mitochondrial calcium (Ca^2+^) homeostasis and activation of caspases.

### 2.7. Cisplatin and Tangeretin Synergistically Enhanced the Cell Cytotoxicity of BFTC-905 Cells

Doxorubicin and cisplatin induced mitochondrial dysfunction in clinical trials. These drugs increased reactive oxygen species (ROS) production and altered mitochondrial enzymatic activity [30]. We used cisplatin and tangeretin to perform cell viability experiments on BFTC-905 cells. Cisplatin and tangeretin synergistically enhanced the ability to inhibit BFTC-905 cell survival, as shown in Figure 7A. Subsequently, we used immunoblotting to analyze the expressions of Bcl-2 family proteins in BFTC-905 cells after treatment with cisplatin and tangeretin. Our results showed that the expressions of Bax and Bad were increased with increasing concentrations of cisplatin and tangeretin treatment, but at the same time, the expressions of Bcl2 and p-Bad were reduced, as shown in Figure 7B. Therefore, the results suggested that cisplatin and tangeretin synergistically caused mitochondrial stress, resulting in an increase in Bad and Bax protein expression and reductions of Bcl2 and p-Bad protein expressions.

## 3. Discussion

### 3.1. Tangeretin Induces Apoptosis in BFTC-905 Cells

Proteomics analysis is a useful tool by which to detect changes in proteins in cells treated with natural compounds, which may help to identify crucial signal transduction pathways of diseases, potentially assisting the development of new feasible therapeutic approaches. In the present study, we investigated the anti-tumor effect of tangeretin on bladder cancer using an in vitro model with a BFTC-905 cell line. Our study demonstrated that tangeretin reduced cell survival by 45% at a concentration of 60 μM, and showed that tangeretin inhibited cell migration in wound-healing and transwell migration assays. The results suggested that tangeretin inhibits bladder cancer cell proliferation and migration.

The occurrence of apoptosis includes several specific indicators, such as cell shrinkage, chromosome condensation, DNA fragmentation, and phosphatidylserine exposure [31]. We found that tangeretin treatment induced DNA fragmentation, which increased the amount of TUNEL dye incorporated in the 3′-hydroxyl termini of the double-strand DNA breaks, and increased TUNEL fluorescent intensity was observed with increasing tangeretin concentrations. As phosphatidylserine exposure occurs in the early stage of apoptosis, it can be used as an indicator of early apoptosis. Flow cytometry analysis showed that tangeretin caused an apoptotic response, as the percentage of early apoptotic cells increased from 1.3 to 10.4%, and that of late apoptotic cells increased from 1.8 to 18% after cells had been incubated with 60 μM tangeretin, as shown in Figure 3.

### 3.2. Tangeretin Induced Mitochondrial Dysfunction, Leading to Apoptosis in BFTC-905 Cells

In order to gain a deeper understanding of the molecular changes in tangeretin-treated BFTC-905 cells, we used 2DGE to identify differentially-expressed proteins regulated by tangeretin. We identified 41 spots that were differentially-expressed on the two-dimensional gel with at least a 1.5-fold increase/decrease in expression between the control cells and the tangeretin-treated cells. The proteins included those involved in mitochondrial energy metabolism, such as ATP synthase D chain, T-complex protein 1, pyruvate kinase isozymes M1/M2, triosephosphate isomerase, and ubiquinol-cytochrome-c reductase complex core protein 2 (UQCRC2). Increased expressions of these proteins indicated that cells may require more energy to initiate the process against mitochondrial stress [32,33]. Among these proteins identified by the proteomics approach, triosephosphate isomerase is a key enzyme that catalyzes gluconeogenesis. Hypoxia may upregulate its expression in cells to gain more energy through glycolysis. UQCRC2 is part of the ubiquinol-cytochrome *C* reductase complex, also known as mitochondrial complex III, which is a subset of mitochondrial respiratory chain complexes. It is speculated that under oxidative stress caused by tangeretin, in order to generate more energy, cells need to increase the UQCRC2 complex expression. The expression of UQCRC2 has been shown to be associated with the anti-cancer effect of HSP60-regulated mitochondrial oxidative phosphorylation [34]. Additionally, the increased expression of peroxiredoxin-6 implied that upregulation of peroxiredoxin may relieve the stress caused by flavonoids. Under stress conditions, cells must obtain a large amount of energy for a series of processes to overcome the stress. The results of our study suggested that tangeretin induced oxidative stress in bladder cancer cells and initiated mitochondrial dysfunction to induce apoptosis.

Many studies have demonstrated that mitochondria are one of the main organelles involved in apoptosis [35,36]. Our study results showed that tangeretin induced apoptosis through mitochondria-mediated pathways in BFTC-905 bladder cancer cells. To explore the mechanism, we used mitochondrial phospholipase inhibitors ArA, CyA, and TFZ, and examined their effects on tangeretin-associated cytotoxicity in BFTC-905 cells. The results showed that the phospholipase inhibitors partially inhibited the cytotoxic effects of tangeretin, as shown in Figure 6. Past studies have reported that abnormal expression of Bcl2 and overexpression of caspase-3 induce apoptosis [37]. Under normal conditions, Bcl-2 family proteins, anti-apoptotic members Bcl2 and Bcl-xL, and pro-apoptotic members Bax and Bad, form heterodimers. Bcl2, Bcl-xL, and p-Bad were down-regulated, while Bad, Bax, and Bid were up-regulated in the tangeretin-treated cells. This suggested that homeostasis destruction of the outer mitochondrial membrane occurred upon tangeretin treatment, resulting in cytochrome *C* release into the cytosol.

In the present study, we validated the anti-tumor effect of tangeretin on BFTC-905 bladder cancer cells. In addition, tangeretin induced the release of pro-apoptotic factors, such as cytochrome *C*, activation of caspase-9 to form apoptotic complexes, and caspase-3 to initiate the apoptotic response. In summary, the current study showed that tangeretin treatment caused cytotoxicity in BFTC-905 cells, which was mediated by mitochondrial dysfunction to trigger apoptotic responses.

## 4. Materials and Methods

### 4.1. Materials

Many reagents, including Dulbecco’s modified Eagle’s medium (DMEM), trypsin-ethylenediaminetetraacetic acid, fetal bovine serum (FBS), and phosphate-buffered saline (PBS), were obtained from Biowest (Nuaillé, France). The 2-D Quant Kit protein assay kit, immobilized pH gradient (IPG) buffer, and isoelectrofocusing strips were obtained from GE Healthcare (Buckinghamshire, UK). Polyvinylidene difluoride (PVDF) membranes, and goat anti-rabbit and horseradish peroxidase-conjugated immunoglobulin (Ig) G were obtained from Millipore (Billerica, MA, USA). Protease inhibitor cocktail, DMSO, ArA, CyA, TFZ were obtained from BioSource International (Camarillo, CA, USA). Cell extraction RIPA buffer was obtained from TOOLS (TOOLS, Taiwan). Enhanced chemiluminescence (ECL) western blotting reagents were obtained from Pierce Biotechnology (Rockford, IL, USA). Antibodies against PDIA3, 14-3-3 sigma, T-complex1, Stress-70, Annexin A2, and UQCRC2, were obtained from Epitomics (Burlingame, CA, USA). Antibodies against Endo G and AIF were obtained from ProteinTech Group (Chicago, IL, USA). Antibodies against pro-caspase 3, cleaved-caspase 3, pro-caspase 9, cleaved-caspase 9, cytochrome *C*, Bid, Bax, Bad, p-Bad, Bcl2, and Bcl-xL were obtained from Cell Signaling Technology (Danvers, MA, USA).

### 4.2. Cell Culture and Tangeretin Treatment

BFTC-905, J82, RT4, and T24 Bladder cancer cells were purchased from the Food Industry Research and Development Institute (Hsinchu, Taiwan) and were grown in DMEM (Biowest, Nuaillé, France), 4 mM L-glutamine, 10% (*v*/*v*) fetal bovine serum, 100 μg/mL streptomycin, 100 U/mL penicillin, and 1 mM sodium pyruvate in a humidified atmosphere of 5% CO_2_ in air at 37 °C. Cells were treated with various concentrations of tangeretin (0~100 μM) and harvested after 24 h of incubation, and the mitochondrial permeability transition inhibitors, 0.5 μM CyA, 25 μM ArA, and 0.5 μM TFZ, were used to pretreat cells 2 h before tangeretin.

### 4.3. Cell MTT Assay

The cell viability effect of tangeretin against BFTC-905, J82, RT4, and T24 cells was examined by calorimetric tetrazolium (MTT) assay. Briefly, BFTC-905, J82, RT4, and T24 cells were seeded in 96-well plates at a density of 1 × 10^5^ in complete medium (with 10% fetal bovine serum) and treated with different concentrations of tangeretin (20, 40, and 60 μM) for 24 h. Cells treated with DMSO without tangeretin were used as a blank control. After incubation, cells were washed and 50 μL MTT solutions added (1 mg/mL in phosphate buffered saline (PBS) buffer) at 37 °C for 4 h. Then, cells were lysed with 200 μL DMSO. The absorbance was determined at 595 nm on a microtiter plate ELISA reader (Bio-Rad, Hercules, CA, USA) with DMSO as the blank. All experiments were carried out in triplicate to confirm the reproducibility [38].

### 4.4. Quantitative Detection of Apoptosis by Flow Cytometry

To determine the apoptosis induced by tangeretin in BFTC-905 cells, an annexin V-FITC/PI Apoptosis Detection kit (Pharmingen, San Diego, CA, USA) was used and the method was as according to a previous study. A total of 1 × 10^6^ cells was seeded onto a 5 cm Petri dish and treated with or without tangeretin for 12 h, and subsequently cells were stained with annexin V-FITC and propidium iodide (PI) for 30 min at 37 °C according to the manufacturer’s protocol. Apoptotic cells were then assessed using a FACScan flow cytometer and Cell-Quest software (Becton-Dickinson, Mansfield, MA, USA) [39].

### 4.5. Wound-Healing and Transwell Migration Assays

For the wound-healing assay, BFTC-905 cells were seeded in six-well plates (8 × 10^5^/well). After the cells reached confluence, a scratch or wound was made with a pipette tip in each well. Unattached tumor cells were washed with PBS and refreshed with FBS-containing medium. Images of the control and experimental groups (0 and 60 μM tangeretin) were acquired at 0, 12, and 24 h after treatment. For the migration assay, 1 × 10^4^ BFTC-905 cells were seeded on a Boyden chamber (Neuro Probe, Cabin John, MD, USA) and then treated with different concentrations of tangeretin (0, 20, 40, 60 μM). After 24 h, the migrated cell stained with 0.1% crystal violet to dye transfer cells on the transwell membrane [40].

### 4.6. DAPI and TUNEL Stain

BFTC-905 cells (1 × 10^5^ cells/well) in a 12-well plate were treated with 20, 40, and 60 μM tangeretin for 24 h, and DMSO was added as the control. Cells in each treatment and control group were fixed with 4% paraformaldehyde in PBS solution for 15 min and stained by DAPI according to the manufacturer’s instructions. The DeadEnd™ Fluorometric TUNEL System (Promega, USA) was used to detect nuclear DNA fragmentation according to the manufacturer’s manual. The cells were photographed under a fluorescence microscope [41].

### 4.7. Protein Preparation and Measurement

BFTC-905 cells were treated with different concentrations of tangeretin (0, 60 μM) for 24 h, and then lysed with the protease inhibitor cocktail and cell extraction buffer. All proteins in the supernatant were then precipitated overnight (−20 °C) using triple volume of 10% trichloroacetic acid/acetone solution containing 20 mM dithiothreitol (DTT) [42]. The pellets were collected and resuspended overnight in a rehydration buffer (6 M urea, 2 M thiourea, 0.5% IPG buffer, 20 mM DTT, 0.5% 3-[(3-cholamidopropyl) dimethylammonio]-1-propanesulfonate (CHAPS), and 0.002% bromophenol blue) at 4 °C. The protein concentration was determined using the 2-D Quant Kit (GE Healthcare, Buckinghamshire, UK).

### 4.8. Two-Dimensional Gel Electrophoresis and Protein Identification by Liquid Chromatography–Tandem Mass Spectrometry

In this study, 2DE was performed with the GE Healthcare Ettan IPGphor 3 and SE 600 Ruby electrophoresis unit (Hoefer, Holliston, MA, USA) by using a previously described protocol [43]. A sample was dissolved in the rehydration buffer as described earlier and applied on an IPG strip in a strip holder. Proteins (100 μg) extracted from whole cells were loaded on an 11 cm IPG strip (*pI* 4–7, 3-10NL Immobiline DryStrip), and subsequently separated by SDS-PAGE (12.5%). The protein spots of interest were excised from the 2DGE gels and digestion with trypsin. Peptides were identified by LC–MS/MS analysis using an AB SCIEX QTRAP^®^ 5500Q mass spectrometer (Applied Biosystems, Framingham, CA, USA).

### 4.9. Western Blotting

Mitochondrial and cytosolic cytochrome *C* were separated using a cytochrome *C* releasing apoptosis assay kit (Biovision, Milpitas, CA, USA). Proteins were separated by SDS-PAGE (12.5%) and transferred to PVDF membranes (Millipore) for 1.5 h at 400 mA using a TE 22 transfer tank (Hoefer, Holliston, MA, USA). The membranes were then incubated overnight with different primary antibodies at 4 °C, washed five times with PBST buffer (10 mM NaH_2_PO_4_, 130 mM NaCl, 0.05% Tween-20), and then probed with the secondary antibody (horseradish peroxidase conjugate, 1:5000 in blocking solution) for 1 h. After the membrane washing process, the immunoreactive bands were visualized through chemiluminescence by adding ECL western blotting reagents (Pierce Biotechnology, Rockford, IL, USA). β-actin was used as a loading control [38].

### 4.10. Statistical Analysis

Results were pooled from three independent experiments. Data from the MTT and cell migration assays as well as the flow cytometric analysis were subjected to analysis of variance (ANOVA) and Tukey’s test using SPSS 14.0 software (IBM, Endicott, NY, USA). Results with *p* < 0.05 were considered statistically significant.

## 5. Conclusions

In this study, we demonstrated that tangeretin induced apoptosis in BFTC-905 cells and confirmed that mitochondria were associated with the apoptotic response using a proteomics approach. Our results also established the mechanism of mitochondrial dysfunction caused by tangeretin in BFTC-905 bladder cancer cells, as shown in Figure 8. This was the first study to analyze the protein changes in the cytotoxic effect of tangeretin on bladder cancer cells using 2DGE. Our findings indicated that tangeretin induces apoptosis in BFTC-905 cells through induction of mitochondrial dysfunction, suggesting that tangeretin has the potential to be developed as an anti-cancer drug for the treatment of bladder cancer.

## Figures and Tables

**Figure 1 ijms-20-01017-f001:**
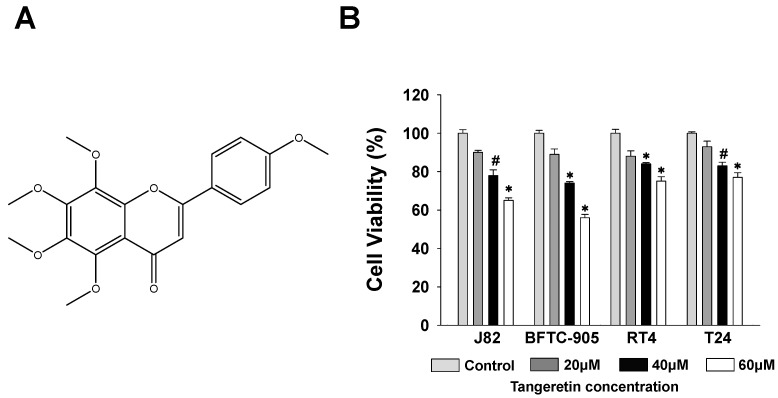
Cell survival of J82, BFTC-905, T24, and RT4 bladder cancer cell lines after tangeretin treatment. (**A**) Chemical structure of tangeretin. (**B**) Cytotoxicity was measured using an MTT assay. # *p* < 0.05, * *p* < 0.001.

**Figure 2 ijms-20-01017-f002:**
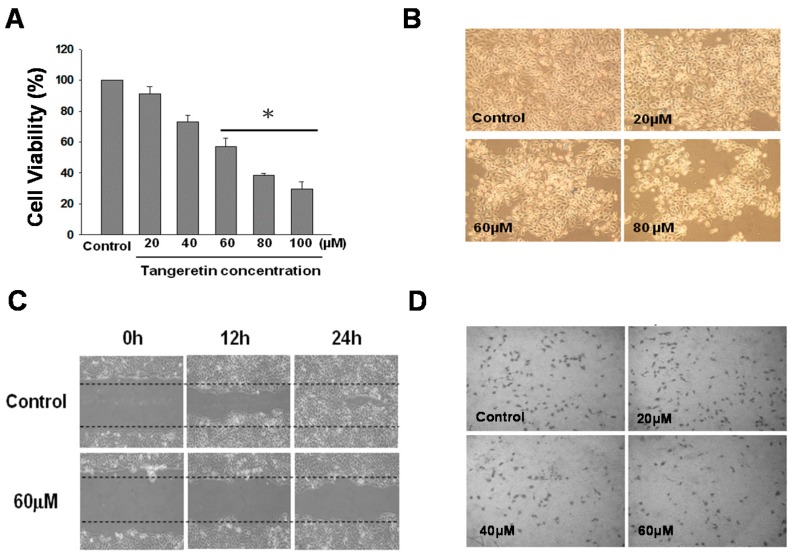
Effect of tangeretin on the cellular behavior of BFTC-905 cells. (100× magnification) (**A**) Effect of tangeretin on cell viability. # *p* < 0.05, * *p* < 0.001. (**B**) Change in cell morphology after tangeretin treatment. (**C**) Effect of tangeretin on wound-healing. (**D**) Effect of tangeretin in a transwell migration assay.

**Figure 3 ijms-20-01017-f003:**
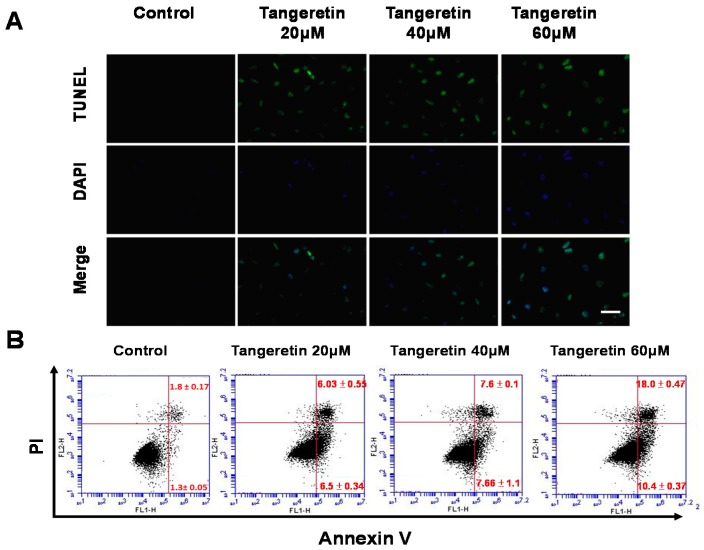
Tangeretin-induced apoptosis in BFTC-905 cells. (**A**) TUNEL/DAPI staining of cells after tangeretin (0, 20, 40, and 60 μM) treatment. Scale bars = 50 μm. (**B**) Annexin V/PI labeling with flow cytometry analysis indicated the percentages of cells in early and late apoptosis after tangeretin treatment.

**Figure 4 ijms-20-01017-f004:**
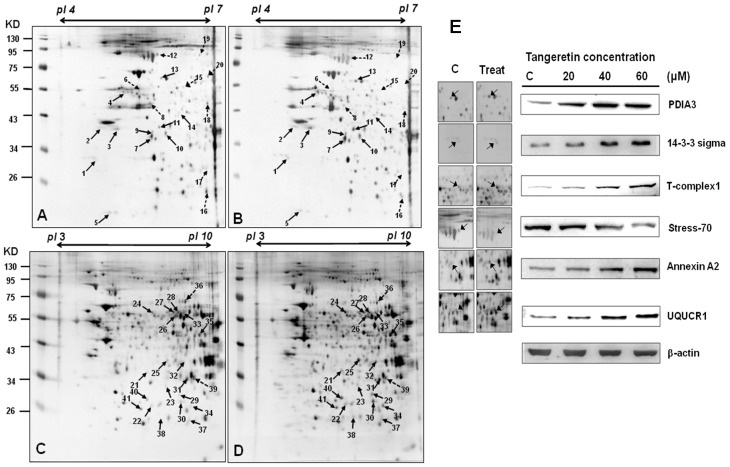
Two-dimensional gel electrophoresis analysis of BFTC-905 cells without and with 60 μM tangeretin treatment. (**A**,**C**) Control, 60 μM tangeretin treatment (**B**,**D**). (**E**) Validation of six differentially-expressed proteins by western blotting from BFTC-905 cells following tangeretin treatment (0, 20, 40, 60 μM).

**Figure 5 ijms-20-01017-f005:**
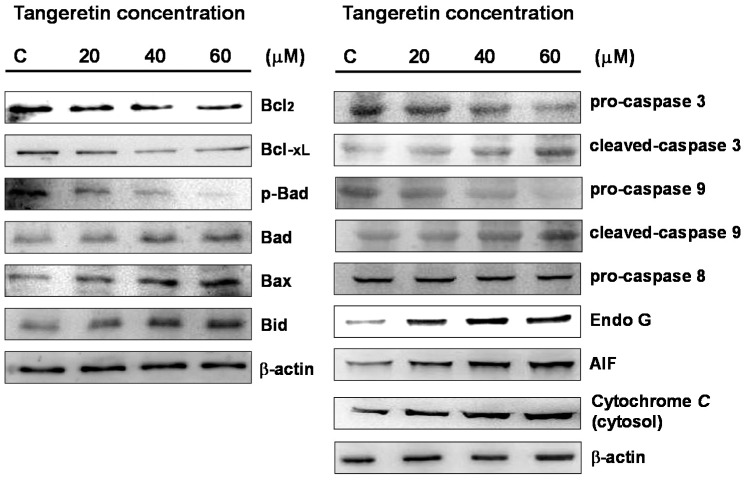
Changes in mitochondrial proteins, Bcl-2 family proteins, and caspases in BFTC-905 cells treated with different concentrations of tangeretin. AIF: apoptosis inducing factor.

**Figure 6 ijms-20-01017-f006:**
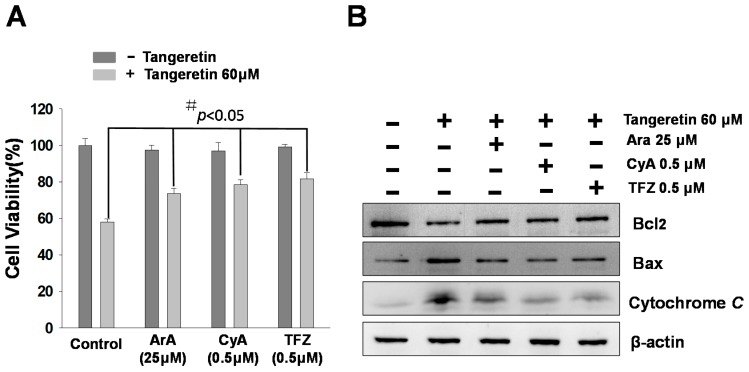
Effect of mitochondrial inhibitors on the cell viability and Bcl-2 family protein expression in tangeretin-treated BFTC-905 cells. (**A**) Cell viability and (**B**) western blotting analysis. Cells were pre-treated with mitochondrial inhibitors trifluoperazine (TFZ), aristolochic acid (ArA), and cyclosporine A (CyA), at the indicated concentrations, followed by 24 h of tangeretin treatment. # *p* < 0.05.

**Figure 7 ijms-20-01017-f007:**
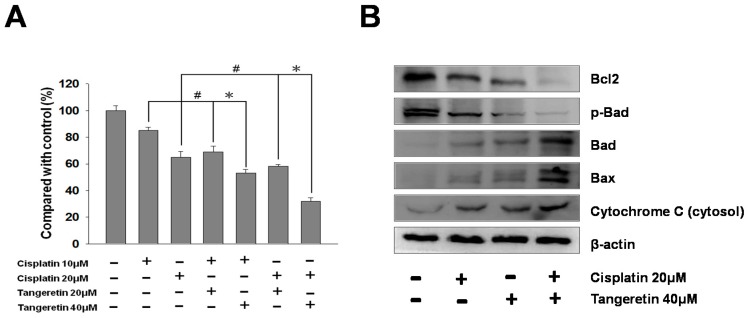
Cell survival of BFTC-905 bladder cancer cells after tangeretin and cisplatin treatment. (**A**) Cytotoxicity was measured using an MTT assay. # *p* < 0.05, * *p* < 0.001. (**B**) Validation of Bcl-2 family proteins by western blotting from BFTC-905 cells following tangeretin and cisplatin treatment.

**Figure 8 ijms-20-01017-f008:**
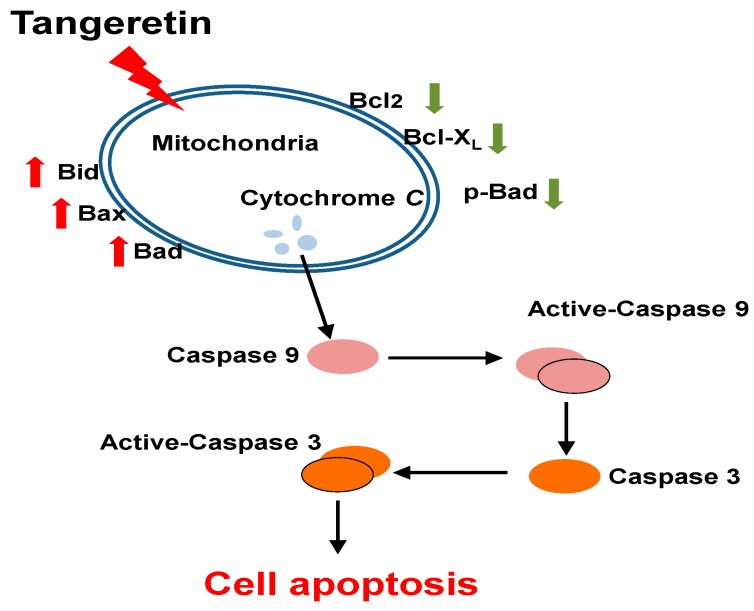
Tangeretin-induced apoptotic pathway in BFTC-905 bladder cancer cells. Based on the results of our study, the apoptosis process caused by tangeretin is mediated by mitochondrial dysfunction and caspase activation.

**Table 1 ijms-20-01017-t001:** Protein identification by LC–MS/MS.

Spot No.	Protein Name	Accession.no	Calculated Mw/pI	Peptide Matched	Sequence Covered %	MASCOT Score	Regulation (Fold- Change) *
1	14-3-3 protein sigma	P31947	27.75/4.68	14	45	158	+2.9
2	Nucleophosmin (NPM)	P06748	32.55/4.64	7	25	87	+3.9
3	Heterogeneous nuclear ribonucleoproteins C1/C2	P07910	33.65/4.95	25	27	301	+3.9
4	Protein disulfide-isomerase A6 precursor	Q15084	48.09/4.95	2	5	77	+4.8
5	ATP synthase D chain, mitochondrial	O75947	18.47/5.21	5	19	84	+2.1
6	Thymidine phosphorylase precursor	P19971	49.92/5.36	14	31	108	−6.8
7	Guanine nucleotide-binding protein subunit beta 4	Q9HAV0	37.54/5.6	3	6	66	+2.5
8	Creatine kinase B-type	P12277	42.61/5.34	12	14	60	−5.9
9	Eukaryotic translation initiation factor 3 subunit 2	Q13347	36.47/5.38	17	39	167	+2.3
10	L-lactate dehydrogenase B chain	P07195	36.61/5.71	19	30	84	+2.2
11	60S acidic ribosomal protein P0 (L10E)	P05388	34.25/5.71	7	26	52	+4.2
12	Stress-70 protein, mitochondrial precursor	P38646	73.63/5.87	11	17	158	−2.8
13	Protein disulfide-isomerase A3 precursor	P30101	56.74/5.98	32	41	270	+2.9
14	DnaJ homolog subfamily B member 11 precursor	Q9UBS4	40.48/5.81	13	26	150	+2.2
15	Heterogeneous nuclear ribonucleoprotein H	P31943	49.19/5.89	15	11	136	−2.6
16	Dermcidin precursor (Preproteolysin)	P81605	11.27/6.08	10	1	43	−2.6
17	Triosephosphate isomerase	P60174	26.65/6.45	23	11	82	+2.1
18	Serpin B3	P29508	44.53/6.35	37	13	176	−3.0
19	Serum albumin precursor	P02768	69.31/5.92	25	21	289	−2.5
20	D-3-phosphoglycerate dehydrogenase	O43175	56.61/6.29	22	14	234	−3.1
21	26S proteasome non-ATPase regulatory subunit 14	O00487	34.55/6.06	3	8	57	+2.7
22	Peroxiredoxin-6	P30041	25.01/6.0	10	34	70	+2.1
23	Purine nucleoside phosphorylase	P00491	32.09/6.45	1	5	89	+2.4
24	T-complex protein 1 subunit beta	P78371	57.45/6.01	8	15	64	+2.7
25	Annexin A1	P04083	38.69/6.57	23	54	392	+3.4
26	Dihydrolipoyl dehydrogenase	P09622	54.11/7.59	8	13	144	+1.8
27	Inosine-5’-monophosphate dehydrogenase 2	P12268	55.77/6.44	6	9	135	+1.7
28	Adenylyl cyclase-associated protein 1	Q01518	51.82/8.27	13	20	140	+1.8
29	Phosphoglycerate mutase 1	P18669	28.78/6.67	11	39	89	+2.5
30	Triosephosphate isomerase	P60174	26.65/6.45	34	62	338	+2.8
31	Electron transfer flavoprotein subunit alpha	P13804	35.05/8.62	12	31	218	+1.9
32	Annexin A2	P07355	38.58/7.57	9	25	133	+2.3
33	Adenylyl cyclase-associated protein 1	Q01518	51.82/8.27	13	13	112	+2.1
34	3-hydroxyacyl-CoA dehydrogenase type-2	Q99714	26.9/7.66	21	60	290	+2.9
35	Ubiquinol-cytochrome-c reductase complex core protein 2	P22695	48.41/8.74	22	28	222	+3.4
36	Heterogeneous nuclear ribonucleoprotein L	P14866	60.14/6.65	21	23	202	−1.8
37	Pyruvate kinase isozymes M1/M2	P14618	57.9/7.96	13	21	178	+2.4
38	Acyl-protein thioesterase 1	O75608	24.65/6.29	7	20	75	+2.6
39	Guanine nucleotide-binding protein subunit beta 2-like 1	P63244	35.05/7.6	8	15	89	−2.2
40	Endoplasmic reticulum protein ERp29 precursor	P30040	28.97/6.77	21	46	264	+4.1
41	Enoyl-CoA hydratase	P30084	31.36/8.34	19	32	302	+3.2

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
