# Peer review of "Proteomics Analysis of Tangeretin-Induced Apoptosis through Mitochondrial Dysfunction in Bladder Cancer Cells"

_ijms, 2019, doi:10.3390/ijms20051017_

Reviewer 1 Report

The authors have addressed all my comments. Only moderate English changes are needed in the Material&Methods section.

Author Response

Reviewer 1

The authors have addressed all my comments. Only moderate English changes are needed in the Material&Methods section.

Respond: Thank reviewer’s suggestion. We have modified material & methods.

Reviewer 2 Report

The paper “Proteomics Analysis of Tangeretin-induced Apoptosis through Mitochondrial Dysfunction in Bladder Cancer Cells” aimed to explore the effect of tangeretin as an anticancer drug for the treatment of bladder cancer. The rationale of the study is interesting, several techniques were used; however, there are several issues that are not clear and some conclusions too speculative.

-          This study has a very long-term goal since no in vivo studies were performed neither the effect of tangeretin was analyzed in cell lines that mimics healthy ones.

-          The cell line chosen was the one more sensitive to tangeretin but no association to disease phenotype was made by the authors.

-          No integrated analysis of proteome data considering the biological processes modulated by tangeretin was performed, it seems that proteome analysis was based on specific proteins.

-          How did authors manage the contribution of FBS from cell culture to BFTC-905 cells proteome?

-          Regarding figure 4F, no statistical differences were noticed among tangeretin treatments?

-          There are some conclusions retrieved from proteome data analysis that are too speculative, for instance, how can authors say that cyt C increased in the cytosol if they analysed whole cell extracts? Emphasis is giving to mitochondrial alterations based on changes is mitochondrial protein levels. However, mitochondrial functionality or biogenesis was no evaluated.

-          In the material and methods section, some references are missing, for instance in subsection 4.4.

-          What was the n used for western blotting analysis? The beta-actin signal in WB is too saturated.

-          There are some spelling errors along the document.

Author Response

eviewer 2

The paper “Proteomics Analysis of Tangeretin-induced Apoptosis through Mitochondrial Dysfunction in Bladder Cancer Cells” aimed to explore the effect of tangeretin as an anticancer drug for the treatment of bladder cancer. The rationale of the study is interesting, several techniques were used; however, there are several issues that are not clear and some conclusions too speculative.

-          This study has a very long-term goal since no in vivo studies were performed neither the effect of tangeretin was analyzed in cell lines that mimics healthy ones.

Respond: Thank reviewer’s suggestion. In this study, we were demonstrated that tangeretin has inhibitory effect on bladder cancer in vitro mechanism. In the future study, we will study in vivo study.

-          The cell line chosen was the one more sensitive to tangeretin but no association to disease phenotype was made by the authors.

Respond: Thank reviewer’s suggestion. In the future study, we will use other cell lines for experiments.

-          No integrated analysis of proteome data considering the biological processes modulated by tangeretin was performed, it seems that proteome analysis was based on specific proteins.

Respond: In this study, two-dimensional electrophoresis was used to analyze differential proteins. It was speculated that functional proteins could be classified with mitochondrial oxidative stress, but it was found that other signals pathways (exp: ER stress). In the study, we were focused in tangeretin-induced apoptosis through mitochondrial dysfunction. In the future, we will study another signal pathway. 

-          How did authors manage the contribution of FBS from cell culture to BFTC-905 cells proteome?

Respond: FBS is an indispensable nutrient in cell culture. We washed the cells 3-5 times with PBS buffer before extracting the cellular proteins to avoid interference with FBS proteins in proteomic analysis.

-          Regarding figure 4F, no statistical differences were noticed among tangeretin treatments?

Respond: Thank reviewer’s suggestion. We have deleted the Figure 4F.

-          There are some conclusions retrieved from proteome data analysis that are too speculative, for instance, how can authors say that cyt C increased in the cytosol if they analysed whole cell extracts? Emphasis is giving to mitochondrial alterations based on changes is mitochondrial protein levels. However, mitochondrial functionality or biogenesis was no evaluated.

Respond: Thank reviewer’s suggestion. This was our mistake. We used the Mitochondrial and cytosolic cytochrome C were separated using a cytochrome C releasing apoptosis assay kit (Biovision, Milpitas, CA, USA) and added in the material and methods 4.9. To verify that tangeretin-induced apoptosis is mediated by mitochondrial inactivation, we pre-treated BFTC-905 cells for 2hr with mitochondrial inhibitors (including trifluoperazine (TFZ), aristolochic acid (ArA) and cyclosporine A (CyA)) followed by 24hr of tangeretin treatment. We will use JC-1 dye and Mitochondrial Potential Changes by Flow Cytometry in the future study.

-          In the material and methods section, some references are missing, for instance in subsection 4.4.

Respond: Thank reviewer’s suggestion. We have modified our mistake.

-          What was the n used for western blotting analysis? The beta-actin signal in WB is too saturated.

Respond: Thank reviewer’s suggestion. We will pay attention to reducing the concentration of b-actin in the future.

-          There are some spelling errors along the document.

Respond: Thank reviewer’s suggestion. We have modified our mistake.

This manuscript is a resubmission of an earlier submission. The following is a list of the peer review reports and author responses from that submission.

Round  1

Reviewer 1 Report

The evaluation of anticancer activity endowed by natural compounds is very interesting, but the authors limit their investigation only against four cell lines, where th tangeretin seems to show modarate/low antiproliferative activity. Why did you select only these kinde of cell lines? Furthermore, sevral corrections are needed. Attached you can find the file with some indications. Taking into consideration that the additional assaysmade on mitochondrial activity were performed only whit tangeretin, you must use a known compound to compare its activity. I propose to improve this manuscript.

Author Response

Reviewer 1

The evaluation of anticancer activity endowed by natural compounds is very interesting, but the authors limit their investigation only against four cell lines, where th tangeretin seems to show modarate/low antiproliferative activity. Why did you select only these kinde of cell lines? Furthermore, sevral corrections are needed. Attached you can find the file with some indications. Taking into consideration that the additional assaysmade on mitochondrial activity were performed only whit tangeretin, you must use a known compound to compare its activity. I propose to improve this manuscript.

Response:

At present, there is still no study on tangeretin in inhibiting bladder cancer proliferation and induce cell apoptosis. Therefore, we first investigated the cytotoxic activity of tangeretin against four bladder cancer cell lines, J82, BFTC-905, T24 and RT4. As BFTC-905 cells are sensitive to tangeretin treatment, they were therefore selected as the test cells for our subsequent experiments. Doxorubicin and Cisplatin were induced mitochondrial dysfunction in clinical. These drug were increased ROS production and altered mitochondrial enzymatic activity. (Ref) We used Doxorubicin, Cisplatin and tangeretin to perform cell viability experiments on BFTC-905 cells. Doxorubicin, Cisplatin and tangeretin were synergistically enhance the ability to inhibit BFTC-905 cells survival. (fig S) Reply due to time issues, we are unable to timely mitochondrial dysfunction associated protein immunostaining analysis. We will add Doxorubicin and Cisplatin to our future experiments

Ref:

Drug induced mitochondrial dysfunction: Mechanisms and adverse clinical consequences. Vuda M, Kamath A. Mitochondrion. 2016 Nov;31:63-74

 Reviewer 2 Report

The manuscript “Proteomics Analysis through Mitochondrial Dysfunction in Bladder Cancer Cells” by Lin and co-workers deals with potential anti-cancer properties of tangeretin. The authors use an established bladder cancer cell line (BFTC-905) to elucidate the potential underlying mechanisms and revealed an induction of apoptosis being mediated through a mitochondrial dysfunction as a potential mode of action.

The manuscript is well written and structured and provides the essential information needed.

Only some minor things need to be addressed by the authors:

-          It would be helpful for the reader to include a figure with the chemical structure of tangeretin.

-          In the Material&Methods section a careful proof-reading concerning the wording needs to be performed.

-          More information regarding the 2D-gel electrophoresis should be provided as well as information on the protein amount used in the different western blots.

-          As the authors compare more than 2 groups a student’s t-test is not the right statistical test. Therefore, the authors should recalculate the statistics by applying an appropriate test.

Author Response

Reviewer 2

The manuscript “Proteomics Analysis through Mitochondrial Dysfunction in Bladder Cancer Cells” by Lin and co-workers deals with potential anti-cancer properties of tangeretin. The authors use an established bladder cancer cell line (BFTC-905) to elucidate the potential underlying mechanisms and revealed an induction of apoptosis being mediated through a mitochondrial dysfunction as a potential mode of action.

The manuscript is well written and structured and provides the essential information needed.

Only some minor things need to be addressed by the authors:

-          It would be helpful for the reader to include a figure with the chemical structure of tangeretin.

-          In the Material&Methods section a careful proof-reading concerning the wording needs to be performed.

Response: Thanks for reviewer’s suggestion. We have added the tangeretin chemical structure in figure 1A.

-          More information regarding the 2D-gel electrophoresis should be provided as well as information on the protein amount used in the different western blots.

Response: Thanks for reviewer’s suggestion. We have added the information in the figure 4F.

-          As the authors compare more than 2 groups a student’s t-test is not the right statistical test. Therefore, the authors should recalculate the statistics by applying an appropriate

Response: Thanks for reviewer’s suggestion. We have modified the mistake in material and methods 4.10.

Round  2

Reviewer 1 Report

I think still that these results are too low to justify the publication in this journal